# Multilocus Genotyping of *Giardia duodenalis* in Mostly Asymptomatic Indigenous People from the Tapirapé Tribe, Brazilian Amazon

**DOI:** 10.3390/pathogens10020206

**Published:** 2021-02-14

**Authors:** Pamela Carolina Köster, Antonio F. Malheiros, Jeffrey J. Shaw, Sooria Balasegaram, Alexander Prendergast, Héloïse Lucaccioni, Luciana Melhorança Moreira, Larissa M. S. Lemos, Alejandro Dashti, Begoña Bailo, Arlei Marcili, Herbert Sousa Soares, Solange Maria Gennari, Rafael Calero-Bernal, David González-Barrio, David Carmena

**Affiliations:** 1Parasitology Reference and Research Laboratory, Spanish National Centre for Microbiology, Majadahonda, 28220 Madrid, Spain; pamelakster@yahoo.com (P.C.K.); dashti.alejandro@gmail.com (A.D.); BEGOBB@isciii.es (B.B.); david.gonzalezb@isciii.es (D.G.-B.); 2Post-Graduation Program in Environmental Science, Faculty of Agricultural and Biological Sciences, University of State of Mato Grosso, Cáceres, MG 78200-000, Brazil; malheiros@unemat.br; 3Department of Biological Sciences, Faculty of Agricultural and Biological Sciences, University of State of Mato Grosso, Cáceres, MG 78200-000, Brazil; luciana.melhoranca@unemat.br; 4Department of Parasitology, Institute of Biomedical Sciences, São Paulo University, São Paulo, SP 05508-000, Brazil; jayusp@hotmail.com; 5Field Epidemiology Services, National Infection Service, Public Health England, London SE1 8UG, UK; Sooria.Balasegaram@phe.gov.uk; 6Independent Researcher, Croydon CR0, UK; akprendergast4@gmail.com; 7European Program for Intervention Epidemiology Training (EPIET), European Centre for Disease Prevention and Control (ECDC), 16973 Stockholm, Sweden; heloiselucaccioni@dgs.min-saude.pt; 8Department of Nursing, Faculty of Health Sciences, University of State of Mato Grosso, Cáceres, MG 78200-000, Brazil; larissa@unemat.br; 9Post-Graduation Program in Veterinary Medicine, Santo Amaro University, São Paulo, SP 04829-300, Brazil; amarcili@usp.br (A.M.); hesoares@prof.unisa.br (H.S.S.); sgennari@usp.br (S.M.G.); 10Department of Preventive Veterinary Medicine and Animal Health, School of Veterinary Medicine, University of São Paulo, São Paulo, SP 05508-270, Brazil; 11SALUVET Research Group, Department of Animal Health, Faculty of Veterinary, Complutense University of Madrid, 28040 Madrid, Spain; r.calero@ucm.es

**Keywords:** *Giardia*, Brazil, Amazon, asymptomatic, community, genotyping, indigenous, risk association, Tapirapé, transmission

## Abstract

Little information is available on the occurrence and genetic variability of the diarrhoea-causing enteric protozoan parasite *Giardia duodenalis* in indigenous communities in Brazil. This cross-sectional epidemiological survey describes the frequency, genotypes, and risk associations for this pathogen in Tapirapé people (Brazilian Amazon) at four sampling campaigns during 2008–2009. Microscopy was used as a screening test, and molecular (PCR and Sanger sequencing) assays targeting the small subunit ribosomal RNA, the glutamate dehydrogenase, the beta-giardin, and the triosephosphate isomerase genes as confirmatory/genotyping methods. Associations between *G. duodenalis* and sociodemographic and clinical variables were investigated using Chi-squared test and univariable/multivariable logistic regression models. Overall, 574 individuals belonging to six tribes participated in the study, with *G. duodenalis* prevalence rates varying from 13.5–21.7%. The infection was positively linked to younger age and tribe. Infected children <15 years old reported more frequent gastrointestinal symptoms compared to adults. Assemblage B accounted for three out of four *G. duodenalis* infections and showed a high genetic diversity. No association between assemblage and age or occurrence of diarrhoea was demonstrated. These data indicate that the most likely source of infection was anthropic and that different pathways (e.g., drinking water) may be involved in the transmission of the parasite.

## 1. Introduction

The flagellated *Giardia duodenalis* (syn. *G. intestinalis*, *G. lamblia*) is a cosmopolitan protozoan parasite that inhabits the gastrointestinal tract of humans and other vertebrate animals. Giardiasis is the most reported intestinal protozoan infection globally, with an estimated 280 million symptomatic cases every year [1]. Asymptomatic infections are even more frequent, both in developing [2,3] and developed [4] countries. Indeed, large epidemiological case-control studies conducted in high-prevalence settings have demonstrated that *G. duodenalis* infection was significantly more common in asymptomatic controls than in cases with diarrhoea [5,6,7]. Host immune status and level of nutrition seem to be key factors in the control of the infection or its progression to active disease [8], although the genotype of the parasite may also play a role in the health/disease balance of the host [9]. When present, clinical manifestations associated with *G. duodenalis* infection may include self-limiting acute diarrhoea, persistent diarrhoea, epigastric pain, nausea, and vomiting [10]. Long-term sequelae, including childhood growth retardation and cognitive impairment, have also been recognised [11,12]. Contrary to severe infections by other diarrhoea-causing protozoan parasites such as *Cryptosporidium* spp. or *Entamoeba histolytica*, giardiasis is rarely fatal and is better considered as a debilitating condition.

Transmission of *G. duodenalis* is through the faecal-oral route, either directly via direct contact with infected humans or animals, or indirectly via ingestion of contaminated food or water. Waterborne transmission is likely the most common source of human infections in poor-resource settings with little or no access to safe drinking water and insufficient sanitary facilities [3]. Because of its strong bond with poverty and elevated socioeconomic impact, giardiasis (together with cryptosporidiosis) joined the Neglected Diseases Initiative launched by the World Health Organisation in 2004 [13].

*Giardia duodenalis* exhibits a considerable degree of genetic heterogeneity, allowing the differentiation of eight (A–H) lineages or assemblages with marked differences in host specificity and range [14]. These genetic variants likely represent cryptic species [15]. Assemblages A and B cause most human infections, but they can also infect other mammalian hosts and are, therefore, considered potentially zoonotic. Assemblages C and D occur mainly in canids, assemblage E in domestic and wild ungulates, assemblage F in cats, assemblage G in rodents, and assemblage H in marine pinnipeds. Human infections by assemblages C–F have been sporadically reported, particularly in children and immunocompromised individuals [14].

A recent review on the epidemiological situation of *G. duodenalis* in Brazil has revealed that this protozoan parasite represents a public health concern in the country, with prevalence rates up to 78% in Minas Gerais State and 70% in São Paulo State in 1998 [16]. Available molecular data in the country have evidenced marked differences in the geographical segregation of *G. duodenalis* assemblages circulating in human populations (Appendix A), domestic and wildlife animal species (Appendix A), surface waters (Appendix A), and fresh produce (Appendix A), likely reflecting disparities in infection sources and transmission pathways. Indeed, contaminated surface waters and having contact with domestic (mainly dog) animals were considered as probable sources of human infections [16]. Despite this relative abundance of epidemiological data, giardiasis has been poorly studied in Brazilian indigenous people, partially due to the geographical isolation and difficulty in accessing these fragile communities. Thus, *G. duodenalis* infections have been documented by conventional (microscopy) methods in the range of 7–47% in the Parakanã indigenous people in the eastern Amazon region [17], in indigenous communities in the municipality of São Gabriel da Cachoeira, Amazonas State [18], in native Brazilian children in the Xingu Indian Reservation, Mato Grosso State [19], in the Maxakali and Xukuru-Kariri indigenous communities, Minas Gerais State [20,21], and in the Terena indigenous people, Mato Grosso do Sul State [22]. However, no information is currently available on the *G. duodenalis* assemblages and sub-assemblages circulating in native Brazilian people. This molecular-based epidemiological survey aims at investigating the genetic diversity of *G. duodenalis* and assessing potential risk and/or protective factors associated with the infection in indigenous people from the Tapirapé tribe living in the Brazilian Amazon.

## 2. Results

### 2.1. Study Population

In this study, a total of 574 individuals (male/female ratio: 0.96; age range: 0.1–88 years old, median: 14.0 years old) of the Tapirapé ethnicity living in six independent tribes (population range: 40–263 inhabitants, standard deviation: 83.3) were censed and invited to participate in four consecutive sampling campaigns during July 2008 and January 2010, in both dry and wet seasons. Overall, 98% (564/574) of the censed individuals participated at least in one sampling campaign. Participation rates ranged from 40% to 93% depending on the tribe and sampling campaign (Table 1). A total of 141 individuals participated in all four sampling campaigns, 201 in three sampling campaigns, 136 in two sampling campaigns, and 86 in a single sampling campaign. The distribution of the participating individuals according to sex, age group, and tribe of origin is also summarised in Table 1. Females (mean: 53.6%, SD: 1.0) participated in the survey more often than males (mean: 46.4%, SD: 1.0). Adults (>15 years old) were the largest group in the surveyed population (mean: 24.6%, SD: 3.2), with children 5 to 14 years of age (mean: 39.4%, SD: 1.9) and children ≤5 years old accounting, in average, for 14.6% (SD: 2.0) of the investigated individuals.

### 2.2. Prevalence of G. duodenalis

Microscopy-based prevalence rates for *G. duodenalis* in the Tapirapé community varied from 13.5% (55/407) in the dry season of 2009 to 21.7% (83/382) in the rainy season of 2010 (Table 2). Over the four sampling campaigns, 35.1% (198/564) individuals tested positive at least once. The occurrence of the parasite was influenced by the seasonality (22% rainy versus 17% dry season, Chi-squared test *p* = 0.022) but not the sampling period (year) (Chi-squared test *p* = 0.126). Subsequent newly diagnosed infections were also more likely to occur in the rainy season (odds ratio: 2.29, 95% CI 1.46–3.68, *p* = 0.0001) although this is dependent on the number of samples analysed. *G. duodenalis* infections were more commonly identified in children aged 0‒4 years old. During the period of study, *G. duodenalis* prevalence varied greatly within and among the six tribes investigated, but tribe 5 presented the highest infection rates in all sampling campaigns.

A total of 43, 17, and 4 individuals tested positive for *G. duodenalis* in two, three, or all four sampling campaigns, respectively (Table 3), although this was dependent on the number of samples. In all cases, children younger than 15 years of age accounted for 50.0% to 88.2% of the subjects where the parasite was detected in two or more sampling campaigns. When considering observations with repeated samples, 61.7% observations were always negative, 4.2% always positive, and 34.1% discontinuously positive. Repeated *G. duodenalis* infections were more frequently detected in the wet season (odds ratio: 1.60, 95% CI 1.12–2.29, *p* = 0.0075) in members of tribe 1 (range: 50.0–58.1%) and, to a lesser extent, in tribe 5 (range: 18.6–50.0%).

### 2.3. Molecular Characterisation of G. duodenalis

The genetic diversity within *G. duodenalis* was investigated in a subset of 70 stool samples from 65 individuals with a positive result for this parasite by conventional microscopy. Five individuals provided stool samples positive to this parasite at two different sampling periods. The presence of the parasite was confirmed by qPCR in 97% (68/70) of these samples. Generated cycle threshold (Ct) values ranged from 18.2 to 35.4 (median: 27.4; SD: 3.7).

Genotyping/sub-genotyping data were obtained for a total of 63 stool samples belonging to 58 individuals (Table 4). Amplification success rates were 100% (63/63) for glutamate dehydrogenase (*gdh*), and 87% (55/63) for beta-giardin *(bg*) and triosephosphate isomerase (*tpi*), respectively. Multilocus sequence typing (MLST) data at the three loci were obtained from 83% (52/63) of the samples. Sequence analyses revealed the presence of assemblages A (25%; 16/63) and B (68%; 43/63). Mixed infections A + B were identified in 6% (4/63) of the samples analysed. No mixed infections involving host-specific assemblages C–H were detected.

Subtyping analyses revealed that sub-assemblage AII (8%, 5/63), mixed AII + AIII infections (8%, 5/63), and ambiguous AII/AIII results (8%, 5/63) were equally distributed within assemblage A. No isolates were identified as sub-assemblage AIII. Within assemblage B, most (50%, 31/63) of the sequences corresponded to ambiguous BIII/BIV results. BIII was identified in 21% (13/63) of the sequences, whereas no isolates belonging to BIV were detected. Out of the four A + B mixed infections detected, one (2%, 1/63) involved sub-assemblages AII + BIII, one (2%, 1/63) sub-assemblages AIII + BIII, and two (3%, 2/63) sub-assemblage AII + B (unknown sub-assemblage). Out of the five individuals with giardiasis at two consecutive sampling periods, three of them (ID: 86, 93, and 282) were infected by BIII/BIV at both sampling periods, indicative of prolonged *G. duodenalis* infection, or re-infection by that very same genotype. In contrast, one individual (ID: 50) was first infected by AII/AIII, and by AIII + B at the following sampling campaign. The remaining individual (ID: 269) was first infected by BIII and then by AII at the following sampling campaign. Both cases were strongly suggestive of re-infection events by different genotypes of the parasite.

### 2.4. Intra-Assemblage Genetic Diversity

Table 5, Table 6 and Table 7, Appendix A show the genetic diversity of the *gdh*, *bg*, and *tpi* representative, partial sequences generated in the present study. These Tables provide information for each sequence including stretch, single nucleotide polymorphisms (SNPs), and GenBank accession number. Assemblage/sub-assemblage assignment was conducted by direct comparison of the sequencing results obtained at the three loci investigated. Sequences presenting double peak positions that could not be unequivocally assigned to a given assemblage/sub-assemblage were reported as ambiguous sequences.

A total of 63 sequences were successfully characterised at the *gdh* locus (Table 5). All 17 assemblage A sequences were unequivocally identified as sub-assemblage AII. Of them, seven sequences were 100% identical to reference sequence L40510. The remaining 10 sequences differed by 1–6 SNPs from L40510. BIII sequences showed a high degree of genetic diversity among them, explaining that 21/24 of the sequences assigned to this sub-assemblage corresponded to distinct genotypes (genetic variants) of the parasite. These sequences differed by 4–13 SNPs from reference sequence AF069059, most of them associated with ambiguous (double peak) positions. Similarly, most (20/22) sequences identified as ambiguous BIII/BIV sequences were different among them, differing by 9–17 SNPs from reference sequence L40508. Virtually all SNPs detected in BIII/IV sequences corresponded to double peaks at single nucleotide positions.

At the *bg* locus, a total of 55 sequences were fully characterised (Table 6). Out of the 14 assemblage A sequences, two belonged to AII and five to AIII. All AII and AIII sequences were identical to reference sequences AY072723 and AY072724, respectively. Five sequences were considered mixed AII + AIII infections based on the presence of two double peak (C415Y and T423Y) positions and taking sequence AY072723 as reference. Two additional sequences corresponded to AII + B and AIII + B mixed infections, differing by 32 and 38 SNPs from reference sequence AY072727, respectively. Except one, all the detected SNPs corresponded to clear double peak positions. Compared to the *gdh* locus, a lower (but still substantial) degree of genetic variability was observed within the 41 sequences assigned to assemblage B at the *bg* locus. All of them differed by 1–6 SNPs from reference sequence AY072727. A genetic variant showing two transitional mutations at positions C165T and A183G was the genotype most frequently detected.

A total of 55 sequences were fully characterised at the *tpi* locus (Table 7). Within assemblage A, 14 sequences were assigned to the sub-assemblage AII. Of them, eight had 100% homology with reference sequence U57897, whereas the remaining six sequences differed by 1–2 SNPs from the latter. Two additional sequences were identified as AII + BIII sequences and presented 94–95 SNPs when aligned with reference sequence U57897. Out of the 25 sequences assigned to BIII, seven showed 100% identity with reference sequence AF069561. The remaining 18 sequences grouped in 16 distinct genotypes that differed by 1–6 SNPs from reference sequence A AF069561. Only a single sequence was confirmed ad BIV, differing by 3 SNPs with reference sequence AF069560. Finally, virtually all (12/13) sequences with a BIII/BIV ambiguous result were different among them, differing from reference sequence AF069560 by 7–11 SNPs. As in the case of the BIII/BIV sequences identified at the *gdh* locus, most of the SNPs identified at the *tpi* locus were associated with ambiguous nucleotide positions.

Figure 1 shows the phylogenetic tree obtained for the *gdh* gene by maximum parsimony and Bayesian methods. All *G. duodenalis* sequences clustered together (monophyletic groups) with different well-supported clades (100% of bootstrap and 1.0 posterior probability). Two major branches were formed and included all (A–F) *G. duodenalis* assemblages. The sequences of indigenous people from the Brazilian Amazon clustered in branches for assemblage A (97% of bootstrap and 1.0 posterior probability) and B (100% of bootstrap and 1.0 posterior probability). In assemblage B, the sequences obtained in this study clustered with sub-assemblages BIII and BIV reference strains. Similar phylogenetic trees for the *bg* and *tpi* sequences generated in the present study are shown in Appendix A, respectively.

### 2.5. Intra-Assemblage B Genetic Diversity Analysis

Genetic diversity was far higher within assemblage B than within assemblage A sequences regardless of the molecular marker used. Multiple sequence alignments of BIII, BIV, and ambiguous BIII/BIV sequences at the *gdh*, *bg*, and *tpi* loci revealed the presence of SNPs in multiple sites across used reference sequences, varying from 11 (for B sequences at the *bg* locus) to 32 (for BIII/BIV sequences at the *tpi* locus) sites (Appendix A). Overall, 611 SNPs were identified among assemblage B sequences in all three loci. Of them, 16.9% (103/611) corresponded to single point mutations, and 83.1% (508/611) to double peaks. Defined positions (hotspots) at each investigated locus tended to accumulate the bulk of these SNPs (66.9%; 409/611). Within *gdh*, C87, T147, G150, C204, C309, and G402 were the main hotspots for BIII sequences (reference sequence: AF069059), and C123, T135, T183, G186, C255, C273, C345, T366, T387, and A438 for BIII/BIV sequences (reference sequence: L40508). Within *tpi*, C108 was the only hotspot for BIII sequences (reference sequence: AF069561), and A5, T57, T131, T134, A176, A395, and C470 for BIII/BIV sequences (reference sequence: AF069560). Within *bg*, the main hotspots identified for B sequences were C165, A183, C309, and T519 (reference sequence: AY072727).

The distribution of single point mutations and double peaks differed substantially among sub-assemblages and loci. At the *gdh* locus, hotspot sites accumulated 57.7% of all SNPs detected in BIII sequences, but this figure increased to 72.9% in BIII/BIV sequences. Double peaks accounted for 37.8% of the SNPs detected in BIII sequences, but for 67.8% of the ambiguous BIII/BIV sequences (Figure 2A). At the *tpi* locus, hotspot sites accumulated 18.2% of all SNPs detected in BIII sequences, but this figure increased to 54.5% in BIII/BIV sequences (Figure 2B). Finally, at the *bg* locus, hotspot sites clustered 78.4% of the SNPs detected in assemblage B sequences, of which 58.1% corresponded to double peaks (Figure 2C).

### 2.6. Risk Association Analysis

#### 2.6.1. Comparing *G. duodenalis* Negative/Ever Positive

Overall, 55% of individuals who tested positive for *G. duodenalis* were female, the median age was 10 years old, with 49% <10 years old. A total of 55% were from tribe 1, followed by 15% from tribe 5. The most frequent clinical signs were normal stool appearance (92%), abdominal pain (53%), and stool consistency type 2 (45%). Overall, 58% did not report hand washing, 84% reported eating with hands, and 18% did not report washing fresh produce. Sanitation was predominantly defecation in the woods (70%) and open defecation near households (21.7%). Microscopy examination also revealed that 53% of individuals were coinfected with *Endolimax nana*, 48% with *Entamoeba coli*, 18% with *Chilomastix mesnili*, 16% with *Ancylostoma* spp., and 11% with *Blastocystis* sp.

Children under <15 years old reported more frequently vomiting, abdominal pain, and abnormal (mucous, bloody, mucous-bloody) faecal appearance compared to adults. However, only differences in abdominal pain appeared significant (Chi-squared test, *p* = 0.016). There were no differences in age or symptoms between the two assemblages A and B (Chi-squared test: age group, *p* = 0.552; faecal consistency, *p* = 0.732; abdominal pain, *p* = 1; vomit, *p* = 0,953).

In univariable analysis, hand washing, older age, tribes 2–4, coinfection with *E. coli*, *E. nana*, and *Iodamoeba* were negatively associated with *G. duodenalis*, while tribe 5, faecal consistency 4, open defecation, and the number of samples were positively associated with *G. duodenalis* (Table 8). Regarding the public health features and symptoms, the multivariable model retained the number of samples, age group, tribe, faecal consistency, faecal appearance, and washing fresh produce. Older age groups had a protective effect (adjusted odds ratio (aOR) = 0.40, 95% CI: 0.20–0.81 in 10–14 years old, and aOR = 0.20 95%, CI: 0.11–0.39 in ≥15 years old, respectively), as tribes 2–3 compared to tribe 1 (aOR = 0.46, 95% CI: 0.24–0.85, aOR = 0.43, 95% CI: 0.21–0.85, aOR = 0.31, 95% CI: 0.14–0.63, respectively). The number of samples was positively associated with higher odds of a *G. duodenalis*-positive result (aOR = 1.46, 95% CI: 1.18–1.81), as washing fresh produce (aOR = 1.95, 95% CI: 1.12–3.44) and faecal consistency type 4 (aOR = 1.84, 95% CI: 1.10–3.37) (Table 9). None of the other pathogens considered was found significantly associated with *G. duodenalis* in the multivariable model with coinfections.

#### 2.6.2. Comparing *G. duodenalis* Serial Results

Out of the 478 observations with more than one sample, 62% (295/478) always tested negative for *G. duodenalis*, 4% (20/478) always tested positive, and 34% (163/478) were discontinuously positive. As such, among observations ever positive for *G. duodenalis*, 11% (20/183) were continuously positive. Of those, 70% were aged 0–9 years old, 55% were from tribe 1, and 35% from tribe 5. Overall, 85% did not report hand washing, 30% not washing fresh produce, 50% reported open defecation near the households, and 45% open defecation in the woods. However, in the multivariable analysis, discontinuous positivity was strongly associated with the number of samples (aOR = 0.30, 95% CI: 0.12–0.66), and tribe 5 (aOR = 4.76, 95% CI: 1.30–18.6), but no further significant association was found (Appendix A).

Finally, comparing observations that were always negative versus always positive, the multivariable model only retained age group and tribe, showing evidence of a protective effect of older age groups (aOR = 0.24, 95% CI: 0.06–0.89 for children 5–9 years old; aOR = 0.17, 95% CI: 0.03–0.68 for children 10–14 years old; aOR = 0.05, 95% CI: 0.01–0.20 for children ≥15 years old), and a strong positive association with tribe 5 (aOR = 5.55, 95% CI: 1.61–19.4) (Appendix A). However, when adjusting for the effect of coinfections; the best fit model suggested a protective effect of *E. nana* (aOR = 0.25, 95% CI: 0.08–73), oldest age group ≥15 years old (aOR = 0.07, 95% CI: 0.01–0.28), and a positive effect of tribe 5 (aOR = 5.87, 95% CI: 1.60–22.1) (Appendix A).

## 3. Discussion

This survey presents new insights into the epidemiology of *G. duodenalis* in Amazonian indigenous communities. The main contributions of the study include the demonstration that (i) giardiasis is a common finding (13–22%) in apparently healthy Tapirapé people, mainly affecting children in the age group of 0–9 years old; (ii) assemblage B was responsible for near 70% of the mostly asymptomatic infections detected; and (iii) a high degree of genetic heterogeneity was observed within assemblage B (but not assemblage A) sequences, regardless of the molecular marker used.

Several epidemiological studies conducted in endemic areas worldwide have shown that *G. duodenalis* infections do not seem to correlate positively with diarrhoea [23,24], demonstrating that asymptomatic giardiasis is the rule rather than the exception in these settings. This fact would explain why giardiasis is systematically absent in global burden estimations of diarrhoeal disease [25]. This seems to be also the case of the present study, where *G. duodenalis* infections were detected similarly in asymptomatic individuals (33.8%) and individuals presenting with diarrhoea or other gastrointestinal manifestations (35.3%). Taken together, this information supports the hypothesis that some enteric protist species (e.g., *Blastocystis* sp., *Dientamoeba fragilis*, *G. duodenalis*) might in fact be protective against disease [26]. This is an attractive possibility implying that these agents are indeed acting as pathobionts (that is, microorganisms that normally live as harmless symbionts but under certain circumstances can be pathogenic) forming part of the host eukaryome.

We have shown in our study that *G. duodenalis* infection was strongly related to younger age and tribe (with tribes 1 and 5 having a higher association) and to seasonality. This may be due to external factors associated with indirect transmission pathways of the infection (e.g., source of drinking water, consumption of contaminated fresh produce, swimming in contaminated surface waters, defecation on the open ground near households, and high density of companion or domestic animals) or increased risk of reinfection within the tribe from other infected members through direct person-to-person contact. Contact with faecally contaminated water and produce may be more likely in the rainy season. Children <15 years old with giardiasis reported more frequently vomiting, abdominal pain, and presence of mucus/blood in faeces compared to adults, although observed differences did not reach statistical significance. Young children with an immature immune system may be at higher risk of infections and probably more severe disease episodes. Thus, older adults may have acquired immunity after a previous infection. Indeed, it has been shown that levels of intestinal inflammation caused by *G. duodenalis* infection decrease with subsequent infections [27,28]. This implies that there is acquired protection against the severity of giardiasis but not from reinfection [29]. In this regard, it should be noted that the composition and abundance of the host’s microbiota have also been suggested to play an important role in the outcome of the infection [30].

Giardiasis was also strongly dependent on the number of samples taken, even considering that conventional microscopy (a method that is largely known to be of limited diagnostic sensitivity) was the screening method for the initial detection of *G. duodenalis* in the present survey. This suggests that possible reinfections or chronic infections with intermittent positivity may be more common than initially anticipated. Reinfection may be more pronounced in the rainy season. In addition, no evident differences between individuals continuously positive/discontinuously positive to *G. duodenalis* were found. However, we should exclude a bias in those presenting for sampling. This is unlikely to be a major factor due to the lack of symptoms in most cases.

Regarding coinfections, the presence of *G. duodenalis* was not associated with any other enteric parasite species, except possibly *E. nana*. These results may be biased by the relatively small number of positive samples detected for certain pathogens and should, therefore, be interpreted with caution. Similarly, a counter-intuitive positive association between *G. duodenalis* with washing fresh produce was found. This result may be the consequence of the potential confounder effect of other variables no considered here such as the manipulation of fresh produce or the use of contaminated washing water. The latter possibility would support the relevance of waterborne transmission for human giardiasis.

Molecular sequence analyses of the three loci used here for genotyping purposes also revealed interesting data. There were no differences in age between individuals infected either by the assemblage A or the assemblage B of *G. duodenalis*. Regarding age-related patterns in the distribution of *G. duodenalis* assemblages, our results are in contrast with those previously obtained in surveys targeting clinical populations. For instance, children have been shown to be more commonly infected by assemblage B (83%, 44/53) than adults (52%, 22/42) in patients of all age groups in Spain [31]. Moreover, in that country, assemblage B was significantly more prevalent than assemblage A in asymptomatic outpatient children, but not in individuals of older age [32].

Remarkably, no association between the occurrence of diarrhoea (or any other gastrointestinal manifestation) and the *G. duodenalis* assemblage involved in the infection was found in the investigated population. This result corroborates that observed in children under 5 years of age (*n* = 222) recruited under the Global Enteric Multicentre Study (GEMS) in Mozambique [33]. However, it should be noted that other surveys have shown different, even contradictory, results. For instance, assemblage A was more prevalent than assemblage B in Bangladeshi people (*n* = 343) [34], in Turkish clinical patients (*n* = 44) [35], and in Spanish outpatient children (*n* = 43) [32]. The opposite trend was reported in asymptomatic infected individuals (*n* = 18) in the Netherlands [36].

Genotyping data generated here demonstrated that assemblage B was responsible for three out of four *G. duodenalis* infections in the Tapirapé people, a similar proportion of that (78%) described in paediatric populations in the Amazonas State [37]. Of note, assemblage A tends to be the predominant *G. duodenalis* genetic variant circulating in humans in Brazil (Appendix A). These facts may be indicative of differences in sources of infection, transmission pathways, or even geographical segregation patterns of the parasite in the country. Lack of non-human, host-specific assemblages C–F seem to suggest that companion, production, and free-living animal species are no significant contributors of giardiasis in the surveyed population. This is in spite of the fact that swine and poultry were reared in all seven tribes, and that domestic dog and cat densities were also high. In addition, cattle (but not sheep) farming was also frequent in the proximity of them. Taking together, these data indicate that human giardiasis is mainly of anthropic nature among the Tapirapé people. The extent and accuracy of this statement should be corroborated in future molecular epidemiological studies including animal and environmental (water) samples.

This study also confirms the high genetic variability within *G. duodenalis* assemblage B (but not assemblage A) reported frequently in similar molecular epidemiological surveys conducted in endemic areas globally [38,39] including Brazil [40,41]. This finding was particularly evident at the *gdh* and *tpi* loci, for which most of the generated BIII (78–87%), BIV (100%), and BIII/BIV (90–92%) sequences corresponded to distinct genotypes of the parasite. Sequences unmistakably assigned to BIII and BIV at the *gdh*/*tpi* loci tended to vary only in one to six positions (hotspots) either as mutations or ambiguous (double peak) sites. In these sets of hotspots, the proportion of sites involving double peaks in BIII sequences varied from 38% at the *gdh* locus to 18% at the *tpi* locus. Interestingly, these percentages increased in both cases to 55–68% in ambiguous BIII/BIV sequences, explaining why these isolates were difficult to allocate to a given sub-assemblage. Two independent mechanisms have been proposed to explain the presence of ambiguous (double peak) positions. The first one involves the occurrence of true mixed infections (e.g., BIII + BIV) and would fit well with an epidemiological scenario characterised by high infection and reinfection rates as the one described in the present study. The second one would be associated with the occurrence of genetic recombination. Evidence for the latter possibility comes from independent investigations demonstrating low levels of allelic sequence heterozygosity (implying a genetic homogenisation mechanism) within assemblage A [42] and, to a lesser extent, within assemblage B [43]. Additional evidence of genetic recombination events has been demonstrated within assemblage B in single (trophozoite and cyst) cells [44] and within sub-assemblages BIII and BIV at the genetic population level [45].

The results obtained in the present study may be biased by certain design and methodological constricts. For instance, the initial screening of *G. duodenalis* was based on conventional microscopy, so the true prevalence of the infection is likely to be underestimated. In addition, there may be a response bias as people may be more or less inclined to return to the study if they had a negative or positive test result. Interestingly, the positivity rate was increased by the number of tests performed, suggesting that over time people were likely to have had a giardiasis episode, that they may have had a false-negative result at microscopy examination, or an inherent response bias in that people who were likely to be positive would return for testing. Limitations associated with the main dataset may arise from the combination of period-specific data, although most of the independent variables considered (e.g., demographics, access to safe drinking water, and sanitary conditions) were not expected to change over time. As our analyses used the first negative test result, we could not further explore the effect of seasonality in the multivariable analysis. However, we have already shown in the descriptive data that seasonality is associated with infections and repeated infections. Lack of association between *G. duodenalis* genetic variants and occurrence of clinical symptoms may be influenced by the fact that other diarrhoea-causing agents (including viral and bacterial pathogens) were not assessed. In addition, suspected mixed infections were not further investigated by cloning of PCR amplicons or next-generation sequencing, methods with high sensitivity able to detect genetic variants of the parasite that are underrepresented in the population pool, and that are otherwise undetectable using conventional PCR methods and Sanger sequencing. Finally, the typing scheme used in the present study may lack enough phylogenetic resolution to correctly differentiate between sub-assemblage BIII and BIV sequences. This issue has been highlighted in recent molecular studies for assemblage B and assemblage A sequences [46,47]. This important point emphasises the need of identifying new markers and of developing novel methods for MLST purposes.

## 4. Materials and Methods 

### 4.1. Study Area

Brazil extends over 8,511,965 km^2^ and includes 724 indigenous lands (ILs) covering a total area of 1,173,770 km^2^ and accounting for 14% of the country’s territory [48]. Most ILs are concentrated in the Legal Amazon, representing 23% of the Amazon territory) [49]. The indigenous people from the Tapirapé ethnicity live in the Serra do Urubu Branco region, Mato Grosso State, a region of tropical forest with typical Amazonian flora and fauna interspersed with clean and closed fields. The Tapirapé exploit this environment alternating agriculture, hunting, gathering, and fishing according to the time of year [49,50]. Farmers´villages have traditionally been in the vicinity of dense forests on high, non-flooding lands. Currently, the Tapirapé ethnic group is made up of approximately 700 individuals living in six tribes with maximum and minimum distances from the main tribe of 70 km and 10 km, respectively. The main tribe is in the municipality of Confresa, Mato Grosso State (Figure 3). Tapirapé people interact frequently with individuals from other ethnic tribes at social events, hunt parties, and other activities.

### 4.2. Sampling and Data Collection

This is a prospective, cross-sectional epidemiological study including four sampling periods covering two dry (July 2008 and July 2009) and two rainy (January 2009 and January 2010) seasons. After obtaining the chief’s (‘cacique’) consent for permission to survey, all members of the tribe were informed about the aim of the project and invited to provide a single stool sample at each of the four scheduled sampling periods. Designated persons in each household were given polystyrene plastic flasks for each member of the household and stool samples were collected on the following day.

Individual standardised questionnaires were completed by a member of our research team in face-to-face interviews with designated persons at sample collection, who provided the requested information for each member of his/her household. Questions included demographics (gender, age, village of origin), clinical manifestations (vomit, abdominal pain), hand and vegetable washing, source of drinking water, use of water treatment, defecation place, and contact with domestic animals and livestock. Provided stool samples were visually inspected for consistency and the presence of mucus or blood. Each participant was assigned a unique distinctive code through the whole period of study, which was used to identify his/her stool sample(s) and associated epidemiological questionnaire(s).

### 4.3. Microscopy Examination

Stool samples were kept at 4 °C before microscopy examination, usually within 48 h of collection. A conventional flotation method using sucrose solution (specific gravity: 1.2 g/cm^3^) was conducted in all stool samples as previously described [51]. Two additional techniques were performed—spontaneous sedimentation [52] and centrifugal-sedimentation in formalin-ether [53]. A sample was considered *G. duodenalis*-positive if cysts of the parasite were detected by at least one of the three methods used. Aliquots of faecal positive samples were stored at –20 °C for downstream molecular analyses. Any other enteric parasite (including helminthic and protist) species found during microscopy observation were also identified and recorded.

### 4.4. DNA Extraction and Purification

Positive stool samples were defrosted and *G. duodenalis* cysts concentrated and purified using the Faust method [54]. Obtained supernatants were subjected to three freeze–thaw cycles to facilitate the mechanical breakage of the cyst wall [55]. Genomic DNA was extracted from the processed supernatants (ca 200 μL) using the PureLink Genomic DNA Mini Kit (Thermo Fisher Scientific, Waltham, MA, USA) according to the manufacturer’s instructions. Extracted and purified DNA samples in molecular grade water (200 μL) were kept at −20 °C and shipped to the Spanish National Centre for Microbiology (Health Institute Carlos III) in Majadahonda (Spain) for downstream genotyping analyses.

### 4.5. Molecular Confirmation of G. duodenalis

Confirmation of *G. duodenalis* infection was achieved using a real-time PCR (qPCR) method targeting a 62-bp region of the gene codifying the small subunit ribosomal RNA (SSU rRNA) of the parasite [56]. Amplification reactions (25 μL) consisted of 3 μL of template DNA, 0.5 µM of each primer Gd-80F and Gd-127R, 0.4 µM of the probe (Appendix A), and 12.5 μL TaqMan^®^ Gene Expression Master Mix (Applied Biosystems, Foster City, CA, USA). Detection of parasitic DNA was performed on a Corbett Rotor GeneTM 6000 real-time PCR system (Qiagen, Hilden, Germany) using an amplification protocol consisting of an initial hold step of 2 min at 55 °C and 15 min at 95 °C, followed by 45 cycles of 15 s at 95 °C and 1 min at 60 °C. Water (no template) and genomic DNA (positive) controls were included in each PCR run.

### 4.6. Molecular Characterisation of G. duodenalis

Giardia duodenalis isolates with a qPCR-positive result were re-assessed by sequence-based multi-locus genotyping of the genes encoding for the glutamate dehydrogenase (gdh), beta-giardin (bg), and triosephosphate isomerase (tpi) proteins of the parasite. A semi-nested PCR was used to amplify a ~432-bp fragment of the gdh gene [57]. PCR reaction mixtures (25 μL) included 5 μL of template DNA and 0.5 μM of the primer pairs GDHeF/GDHiR in the primary reaction and GDHiF/GDHiR in the secondary reaction (Appendix A). Both amplification protocols consisted of an initial denaturation step at 95 °C for 3 min, followed by 35 cycles of 95 °C for 30 s, 55 °C for 30 s, and 72 °C for 1 min, with a final extension of 72 °C for 7 min.

A nested PCR was used to amplify a ~511 bp-fragment of the *bg* gene [58]. PCR reaction mixtures (25 μL) consisted of 3 μL of template DNA and 0.4 μM of the primers sets G7_F/G759_R in the primary reaction and G99_F/G609_R in the secondary reaction (Appendix A). The primary PCR reaction was carried out with the following amplification conditions: one step of 95 °C for 7 min, followed by 35 cycles of 95 °C for 30 s, 65 °C for 30 s, and 72 °C for 1 min, with a final extension of 72 °C for 7 min. The conditions for the secondary PCR were identical to the primary PCR except that the annealing temperature was 55 °C.

A nested PCR was used to amplify a ~530 bp-fragment of the *tpi* gene [59]. PCR reaction mixtures (50 μL) included 2–2.5 μL of template DNA and 0.2 μM of the primer pairs AL3543/AL3546 in the primary reaction and AL3544/AL3545 in the secondary reaction (Appendix A). Both amplification protocols consisted of an initial denaturation step at 94 °C for 5 min, followed by 35 cycles of 94 °C for 45 s, 50 °C for 45 s, and 72 °C for 1 min, with a final extension of 72 °C for 10 min.

The semi-nested and nested PCR protocols described above were conducted on a 2720 Thermal Cycler (Applied Biosystems). Reaction mixes always included 2.5 units of MyTAQ^TM^ DNA polymerase (Bioline GmbH, Luckenwalde, Germany), and 5× MyTAQTM reaction buffer containing 5 mM dNTPs and 15 mM MgCl_2_. Laboratory-confirmed positive and negative DNA samples for each parasite species investigated were routinely used as controls and included in each round of PCR. PCR amplicons were visualised on 2% D5 agarose gels (Conda, Madrid, Spain) stained with Pronasafe nucleic acid staining solution (Conda). Positive PCR products were directly sequenced in both directions using appropriate internal primer sets (Appendix A). DNA sequencing was conducted by capillary electrophoresis using BigDye^®^ Terminator chemistry (Applied Biosystems) on an on ABI PRISM 3130 Genetic Analyser.

### 4.7. Sequence and Phylogenetic Analyses

Raw sequencing data in both forward and reverse directions were viewed using the Chromas Lite version 2.1 sequence analysis program (https://technelysium.com.au/wp/chromas/ (accessed on 1 February 2021)). The Basic Local Alignment Search Tool (BLAST) (http://blast.ncbi.nlm.nih.gov/Blast.cgi (accessed on 1 February 2021)) was used to compare nucleotide sequences with sequences retrieved from the NCBI GenBank database. Generated DNA consensus sequences were aligned to appropriate reference sequences using the MEGA 6free software [60] for species confirmation and assemblage/sub-assemblage identification.

For the estimation of the phylogenetic relationships among the identified Giardia-positive samples, gdh sequences generated in this study and human- and animal-derived homologue sequences mostly from Brazil retrieved from GenBank were aligned using Clustal X and adjusted manually with GeneDoc [61,62]. Inferences by maximum parsimony (MP) were constructed by PAUP version 4.0b10 using a heuristic search in 1000 replicates, 500 bootstrap replicates, random stepwise addition starting trees (with random addition sequences), and tree bisection and reconnection branch swapping [63]. MrBayes v3.1.2 was used to perform Bayesian analyses with four independent Markov chain runs for 1,000,000 metropolis-coupled MCMC generations, sampling a tree every 100th generation [64]. References [65,66,67,68,69,70,71,72,73,74,75,76,77,78,79,80,81,82,83,84,85,86,87,88,89,90,91,92,93,94,95,96,97,98,99,100,101,102,103,104,105,106,107,108,109] are cited in the Appendix A. The first 25% of trees represented burn-in and the remaining trees were used to calculate Bayesian posterior probability. The GTR +I + G substitution model was used. The gdh sequence of G. ardeae was used as the outgroup.

The sequences obtained in this study have been deposited in GenBank under accession numbers MT542718-MT542765 (gdh), MT542766-MT542794 (bg), and MT542795-MT542829 (tpi).

### 4.8. Statistical Analysis

We investigated factors (public health features, clinical symptoms, coinfection with other pathogens) associated with a positive *G. duodenalis* result. The main dataset was constructed with data from one of the four sampling points—if the observation ever tested positive for *G. duodenalis*, we used data from the sampling point of the first positive *G. duodenalis* result; otherwise, we used data from the first sampling point in order.

We conducted Chi-squared tests (*p* < 0.05) to compare characteristics of cases and non-cases, and we calculated crude odds ratios (OR) with 95% confidence intervals (CI) to investigate the crude association between independent variables and a *G. duodenalis*-positive result. We constructed multivariable logistic regression models to assess the association between *G. duodenalis* and (i) public health features and clinical signs or (ii) coinfection with other intestinal pathogens, adjusted by age, tribe, and the number of samples. Additionally, we considered the serial results of *G. duodenalis* for observations with at least two samples. We conducted similar analyses by comparing those continuously negative versus continuously positive, and those discontinuously positive versus continuously positive.

Univariable analyses were conducted on all available observations, but observations with missing values were removed from multivariable analyses. All the independent variables were included in the analyses and we used the stepwise backward selection method, removing successively the least significant variable and using Akaike information criterion (AIC) and Bayesian information criterion (BIC) to construct the best fit model. Analyses were performed in R (package stats).

### 4.9. Ethics Approval

This study has been approved by the National Research Ethics Commission (CONEP), Ministry of Health (Brazil), under reference number 120/2008.

## 5. Conclusions

This microscopy-based survey demonstrates that symptomatic and asymptomatic giardiasis are common in indigenous people from the Brazilian Amazon. Children under 15 years of age were particularly exposed to the infection, suggesting that acquired immunity plays a role in modulating the frequency and virulence of the disease. *G. duodenalis* infection rates varied largely among the surveyed tribes and sampling periods, suggesting that different pathways may be involved in the transmission of the parasite. Molecular sequence data indicated that the most likely source of infection was anthropic. The distribution of assemblages was independent of the occurrence of clinical manifestations, indicating that the genotype of the parasite was not associated with the outcome of the infection. Assemblage B accounted for near 75% of the infections detected and showed a high genetic diversity that impaired the correct identification of sub-assemblages BIII and BIV. This diversity was mainly associated with the presence of ambiguous positions (double peaks) at the chromatogram level, suggesting that coinfections and/or genetic recombination events were taking place, at unknown rates, in the investigated population. Further molecular epidemiological studies targeting animal (including domestic and wildlife) and environmental (drinking water) samples are needed to elucidate the transmission dynamics of *G. duodenalis* in this Brazilian geographical region.

## Figures and Tables

**Figure 1 pathogens-10-00206-f001:**
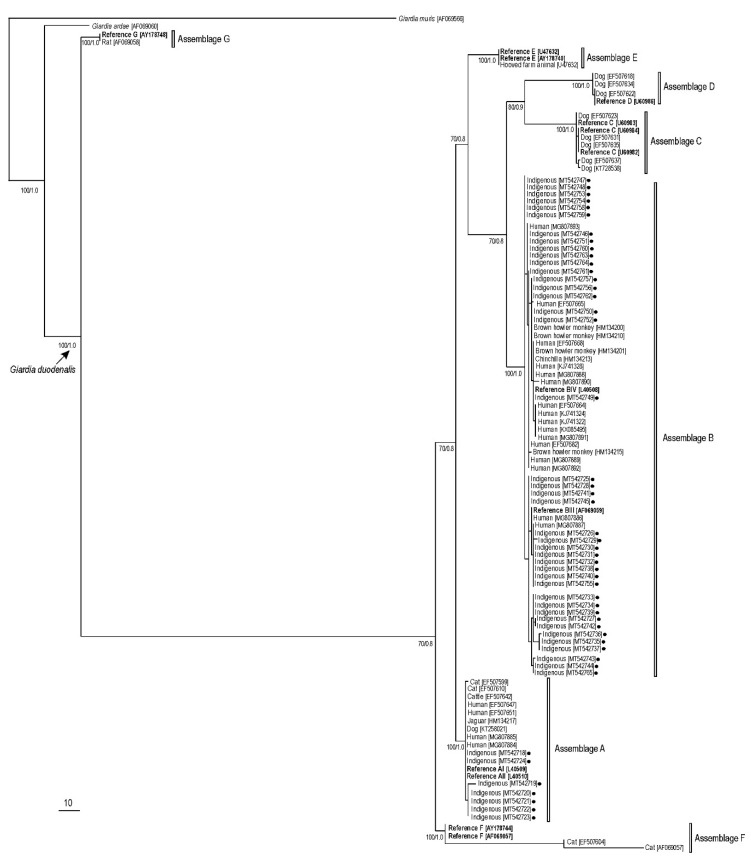
Maximum parsimony phylogenetic tree based on *gdh* sequences of *G. duodenalis*. Numbers on nodes indicate the bootstrap/posterior probability values. Black filled circles represent sequences generated in the present study. GenBank accession numbers for all sequences used for the phylogenetic analysis were embedded in the tree. *Giardia muris* was used as the outgroup.

**Figure 2 pathogens-10-00206-f002:**
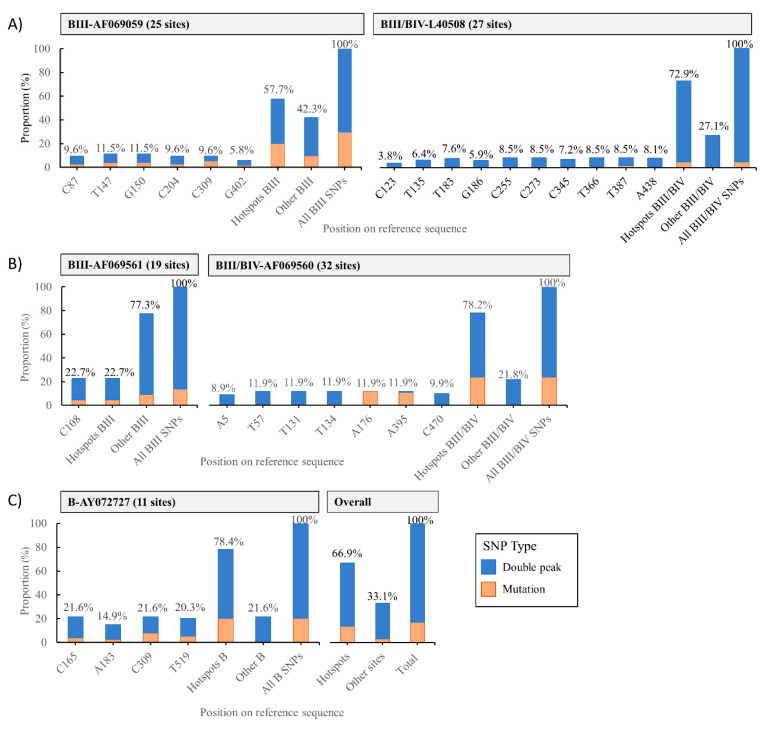
Distribution of single nucleotide polymorphisms segregated by mutations and double peaks, in *G. duodenalis* assemblage B sequences. (**A**) Single nucleotide polymorphisms (SNPs) at the glutamate dehydrogenase (*gdh*) locus; (**B**) SNPs at the triosephosphate isomerase (*tpi*); (**C**) SNPs at the beta-giardin (*bg*) locus, and overall figures for all assemblage B sequences.

**Figure 3 pathogens-10-00206-f003:**
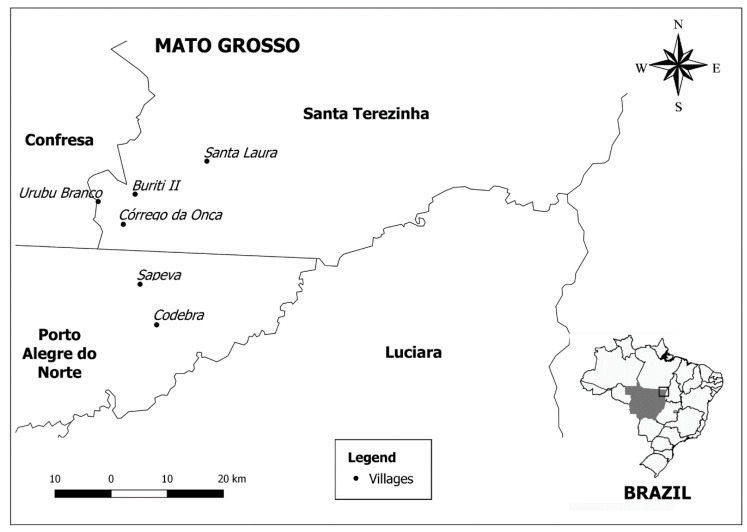
Map showing the exact geographical location of the sites sampled in the present study.

**Table 1 pathogens-10-00206-t001:** Participation rates and distribution by sex, age group, and tribe of origin of the Tapirapé people (*n* = 564) taking part in the four sampling campaigns conducted in the present survey, Brazilian Amazon.

	Participation		Sex		Age group (years)					Tribe *n* (%)					
Sampling Campaign	Participants (*n*)	Rate (%)	Male	Female	0–4	5–9	10–14	≥15	Unknown	1	2	3	4	5	6
2008 ^1^	362	64	171	191	45	73	74	170	0	175 (66.5)	31 (40.3)	46 (65.7)	47 (63.5)	28 (56.0)	35 (87.5)
2009 ^2^	374	66	175	199	53	75	66	178	2	174 (66.2)	45 (58.4)	31 (44.3)	41 (55.4)	46 (92.0)	37 (92.5)
2009 ^1^	407	72	189	218	59	90	64	192	2	202 (76.8)	49 (63.6)	36 (51.4)	42 (56.8)	45 (90.0)	33 (82.5)
2010 ^2^	382	68	172	210	66	94	64	156	2	193 (73.4)	51 (66.2)	38 (54.3)	35 (47.3)	39 (78.0)	26 (65.0)

^1^ Dry season. ^2^ Rainy season.

**Table 2 pathogens-10-00206-t002:** Microscopy-based prevalence of *Giardia duodenalis* by sex, age group, and tribe of origin of the Tapirapé people (*n* = 564) participating in the present survey according to the sampling period, Brazilian Amazon.

	2008 ^1^		2009 ^2^		2009 ^1^		2010 ^2^	
Variable	*Giardia* Positive	%	*Giardia* Positive	%	*Giardia* Positive	%	*Giardia* Positive	%
Sex								
Male	31	19	35	20	30	16	39	23
Female	37	18	46	23	25	12	43	21
Age group (years)								
0–4	13	29	17	32	17	29	28	42
5–9	17	23	20	27	15	17	25	27
10–14	16	22	14	21	9	14	11	17
≥15	22	13	30	17	14	7	19	12
Tribe								
1	41	23	34	20	30	15	48	25
2	2	7	10	22	7	14	6	12
3	4	9	3	10	2	6	6	16
4	4	9	2	5	1	2	5	14
5	10	36	21	46	13	29	14	36
6	7	20	11	30	2	6	4	15
Total	68	19	81	22	55	14	83	22

^1^ Dry season. ^2^ Rainy season.

**Table 3 pathogens-10-00206-t003:** Number of individuals with a positive result to *Giardia duodenalis* by microscopy in two or more of the sampling campaigns conducted in the present study, Brazilian Amazon.

Variable	Positives in 2 Campaigns (*n*)	Frequency (%)	Positives in 3 Campaigns (*n*)	Frequency (%)	Positives in all 4 Campaigns (*n*)	Frequency (%)
Age group (years)						
0–4	14	33	6	35	1	25
5–9	12	28	6	35	1	25
10–14	9	21	3	18	0	0
≥15	8	19	2	12	2	50
Total	43	100	17	100	4	100
Tribe						
1	25	58	7	41	2	50
2	4	9	1	6	0	0
3	1	2	0	0	0	0
4	1	2	0	0	0	0
5	8	19	7	41	2	50
6	4	9	2	12	0	0
Total	43	100	17	100	4	100

**Table 4 pathogens-10-00206-t004:** Multilocus genotyping results of the 63 *G. duodenalis*-positive samples of human origin successfully genotyped at any of the three loci investigated in the present survey.

Sample ID	Ct value in qPCR	*gdh*	*bg*	*tpi*	Assigned Genotype
5	27.4	BIII	B	BIII	BIII
13	25.1	AII	AII + AIII	AII	AII + AIII
24	27.2	AII	AII + AIII	AII	AII + AIII
31	26.4	BIII/BIV	B	BIII/BIV	BIII/BIV
33	28.2	BIII/BIV	B	BIII/BIV	BIII/BIV
41	24.0	BIII	B	BIII/BIV	BIII/BIV
49	25.5	BIII/BIV	B	BIII	BIII/BIV
50a	21.2	AII	AIII	AII	AII/AIII
50b	23.8	BIII	AIII + B	BIII	AIII + BIII
55	27.3	BIII	B	BIV	BIII/BIV
58	29.6	BIII	B	BIII/BIV	BIII/BIV
60	29.1	BIII	B	BIII/BIV	BIII/BIV
70	34.0	BIII	‒	‒	BIII
71	30.0	BIII	B	BIII/BIV	BIII/BIV
72	21.7	BIII	B	BIII/BIV	BIII/BIV
78	18.2	BIII	B	BIII	BIII
79	29.4	BIII/BIV	B	BIII	BIII/BIV
82	30.5	BIII	B	AII + BIII	AII + BIII
85	29.7	BIII/BIV	B	BIII	BIII/BIV
86a	27.0	BIII	B	BIII/BIV	BIII/BIV
86b	26.3	BIII	B	BIII/BIV	BIII/BIV
93a	30.4	BIII/BIV	B	BIII/BIV	BIII/BIV
93b	24.9	BIII/BIV	B	BIII/BIV	BIII/BIV
94	22.0	AII	‒	AII	AII
106	21.3	AII	AIII	AII	AII/AIII
111	26.2	AII	AIII	AII	AII/AIII
123	34.9	BIII	‒	‒	BIII
128	30.5	AII	AIII	AII	AII/AIII
131	23.5	BIII/BIV	B	BIII/BIV	BIII/BIV
132	29.5	BIII	B	BIII	BIII
149	23.6	BIII	B	‒	BIII
157	28.2	BIII/BIV	B	BIII	BIII/BIV
168	24.0	BIII/BIV	B	BIII	BIII/BIV
172	28.3	BIII/BIV	B	BIII	BIII/BIV
176b	29.3	BIII	‒	‒	BIII
179	27.5	AII	B	AII	AII + B
180	23.3	BIII/BIV	B	BIII/BIV	BIII/BIV
182	27.5	BIII/BIV	B	BIII	BIII/BIV
184	31.3	AII	‒	AII	AII
186	26.3	BIII/BIV	B	BIII	BIII/BIV
198	22.6	BIII/BIV	B	BIII	BIII/BIV
200	25.2	BIII/BIV	B	BIII	BIII/BIV
216	25.8	BIII	B	‒	BIII
227	19.5	AII	AII + AIII	AII	AII + AIII
242	28.3	BIII/BIV	B	BIII	BIII/BIV
245	22.4	BIII	B	BIII	BIII
269a	31.2	BIII	‒	BIII	BIII
269b	25.1	AII	AII	AII	AII
282a	27.3	BIII/BIV	B	BIII	BIII/BIV
282b	23.4	BIII/BIV	B	BIII	BIII/BIV
328	32.0	AII	‒	‒	AII
330	30.3	BIII/BIV	B	‒	BIII/BIV
353	25.3	BIII	B	BIII	BIII
374	21.2	BIII/BIV	B	BIII	BIII/BIV
379	23.7	AII	AII + AIII	AII	AII + AIII
420	19.2	AII	AII + AIII	AII	AII + AIII
570	24.7	BIII/BIV	B	BIII	BIII/BIV
572	28.3	AII	AII + B	AII	AII + B
585	24.9	BIII	B	BIII	BIII
595	25.1	AII	AIII	‒	AII/AIII
596	35.4	AII	‒	‒	AII
604	21.7	BIII	B	BIII/BIV	BIII/BIV
607	28.1	BIII	B	BIII	BIII

*bg*, beta-giardin; Ct, cycle threshold; *gdh*, glutamate dehydrogenase; qPCR: real-time polymerase chain reaction; *tpi*, triosephosphate isomerase.

**Table 5 pathogens-10-00206-t005:** Diversity, frequency, and main molecular features of *G. duodenalis* sequences at the *gdh* locus generated in the present study. GenBank accession numbers are provided.

Assemblage	Sub-Assemblage	No. of Isolates	Reference Sequence	Stretch	Single Nucleotide Polymorphisms	GenBank ID
A	AII	7	L40510	64–491	None	MT542718
		1	L40510	66–486	T88C, C128T, C179T, T385C, A410G, T418A	MT542719
		1	L40510	64–491	T132Y, C179T, A224M, G332R	MT542720
		1	L40510	64–491	C150Y, C179T	MT542721
		6	L40510	64–491	C179T	MT542722
		1	L40510	68–491	C179T, T355Y	MT542723
		1	L40510	64–491	C179Y	MT542724
B	BIII	1	AF069059	28–455	C39T, C87Y, C99Y, C123Y, T147Y, G150R, C204Y, C309Y	MT542725
		1	AF069059	30–455	C39T, T219Y, C309T, C351Y	MT542726
		1	AF069059	28–455	C39T, C87Y, C135Y, T147Y, G150R, C204Y, C309T, C336Y	MT542727
		1	AF069059	28–455	C39Y, C99Y, T147Y, G150R, C204Y, C360Y, C387Y, G402R	MT542728
		1	AF069059	28–455	C39T, C309T, G372A, C375T	MT542729
		2	AF069059	40–455	C87Y, C99Y, C123Y, T147Y, G150R, C204Y, C309T	MT542730
		1	AF069059	40–455	C87Y, C99Y, C123Y, T147Y, G150R, C204Y, C309T, G402R	MT542731
		1	AF069059	47–455	C87Y, C99Y, T147Y, G150R, C204Y, T237Y, C309Y, G354R, G372R, C375Y	MT542732
		1	AF069059	47–455	C87Y, C132Y, T147Y, G150R, C174Y, C204Y, T237Y, C309Y, A324R, G354R, G372R, C375Y, G402R	MT542733
		2	AF069059	49–455	C87Y, C132Y, T147Y, G150R, C174Y, C204Y, C309Y, A324R, G372R, C375Y, G402R	MT542734
		1	AF069059	41–460	C87T, C132T, T147C, G150A, C174T, C204T, A324G, G402A	MT542735
		1	AF069059	40–455	C87T, C132Y, T147C, G150A, C174T, C204T, A324R, G402A	MT542736
		1	AF069059	44–443	C87T, C132T, T147C, G150A, C174Y, C204T, G402A, G406A	MT542737
		1	AF069059	46–454	C87Y, C135Y, T147Y, G150R, T209Y, C309T, C336Y	MT542738
		1	AF069059	40–455	C87Y, T147Y, G150R, T237Y, C309Y, G354R, C360Y, C387Y, G402R	MT542739
		2	AF069059	40–455	C87Y, C204Y, C309T, G372R, C375Y	MT542740
		1	AF069059	48–455	C99Y, C135Y, T147Y, G150R, C204Y, C309Y	MT542741
		1	AF069059	82–443	C99T, T147C, G150A, C204T	MT542742
		1	AF069059	44–389	C99T, T147C, G150A, T308Y	MT542743
		1	AF069059	46–455	C87Y, C99Y, C123Y, C132Y, T147Y, G150R, C174Y, C204Y, C309Y, A324R, G402R	MT542744
		1	AF069059	40–455	C87T, T147C, G150A, C309T	MT542745
	BIII/BIV	1	L40508	76–491	G84R, C159Y, T183Y, G186R, C255Y, C273Y, C345Y, T366Y, T387C, G408R, A438G	MT542746
		1	L40508	85–491	C123Y, T135Y, C159Y, T183Y, G186R, C240Y, C255Y, C273Y, C345Y, T366Y, T387Y, A438R	MT542747
		2	L40508	76–491	C123Y, T135Y, C168Y, G180R, T183Y, G186R, C210Y, C240Y, C255Y, C273Y, A360R, T366Y, T387Y	MT542748
		1	L40508	76–491	C123Y, T135Y, C168Y, T183Y, G186R, C210Y, C240Y, C255Y, C273Y, C291Y, C345Y, A360R, T366Y, C372Y, T387Y, C423Y, A438R	MT542749
		1	L40508	76–491	C123Y, T135Y, C168Y, T183Y, G186R, C210Y, C240Y, C255Y, C273Y, C345Y, A360R, T366Y, C372Y, T387Y, C423Y, A438R	MT542750
		1	L40508	101–491	C123Y, T135Y, C168Y, T183Y, G186R, C210Y, C240Y, C255Y, C273Y, C345Y, T366C, T387Y, A438R	MT542751
		2	L40508	77–491	C123Y, T135Y, C171Y, G180R, C240Y, C255Y, C273Y, C345Y, T366Y, T387Y, C423Y, A438R	MT542752
		1	L40508	76–491	C123Y, T135Y, T183Y, C240Y, C255Y, C273Y, C291Y, C345Y, T366Y, C372Y, T387Y, G408R, C411Y, A438R	MT542753
		1	L40508	76–491	C123Y, T135Y, T183Y, C240Y, C255Y, C273Y, C345Y, T366Y, T387Y, G408R, C411Y, A438R	MT542754
		1	L40508	76–491	C123Y, T135Y, C240Y, C255Y, C273Y, C291Y, C345T, T366C, C372Y, T387Y, A438G	MT542755
		1	L40508	76–491	T135Y, C159Y, T183Y, G186R, C240Y, C255Y, C273Y, C345Y, T366Y, T387Y, G408R, C423Y, A438R	MT542756
		1	L40508	84–491	T135Y, C159Y, T183Y, G186R, C240Y, C255Y, C273Y, C345Y, T366Y, T387Y, C423Y, A438R	MT542757
		1	L40508	82–491	T135Y, C159Y, T183Y, G186R, C255Y, C273Y, C345Y, T366Y, T387C, G408R, C411Y, A438R	MT542758
		1	L40508	76–491	T135Y, C159Y, T183Y, G186R, C255Y, C273Y, C345Y, T366Y, T387Y, C396Y, C423Y, A438R	MT542759
		1	L40508	84–491	T135Y, C177Y, T183Y, C255Y, C273Y, C345Y, T366Y, T387Y, G390R, G408R, C411Y, A438R	MT542760
		1	L40508	76–491	T135Y, T183Y, C255Y, C273Y, T366Y, T387C, G408R, A438R	MT542761
		1	L40508	76–491	C159Y, T183Y, G186R, C255Y, C273Y, C345Y, T366Y, T387Y, A438R	MT542762
		1	L40508	76–491	C159Y, T183C, G186R, C255Y, C273Y, C345Y, T366Y, T387Y, C423Y, A438R	MT542763
		1	L40508	76–491	C159Y, T183Y, G186R, C255Y, C273Y, C345Y, T366Y, T387Y, A438R, A476R	MT542764
		1	L40508	76–491	T183C, G186A, C240Y, C255Y, C273Y, T366Y, T387C, A438R	MT542765

M, A/C; R, A/G; Y, C/T.

**Table 6 pathogens-10-00206-t006:** Diversity, frequency, and main molecular features of *G. duodenalis* sequences at the *bg* locus generated in the present study. GenBank accession numbers are provided.

Assemblage	Sub-Assemblage	No. of Isolates	Reference Sequence	Stretch	Single Nucleotide Polymorphisms	GenBank ID
A	AII	2	AY072723	97–719	None	MT542766
	AIII	5	AY072724	1–753	None	MT542767
	AII + AIII	5	AY072723	1–729	C415Y, T423Y	MT542768
	AIII + B	1	AY072727	98–604	T132Y, G159R, A171R, A183R, A228R, T240Y, G258R, C288Y, T306Y, C309Y, T312K, G327R, C339Y, G345R, C348Y, C372Y, A387R, C390Y, C415Y, A432R, C438Y, C453M, G456R, T471Y, A489R, T519Y, C522R, C558Y, C561Y, C564Y, G576R, C579S	MT542769
	AII + B	1	AY072727	96–603	T132Y, G159R, C165Y, A171R, A183R, A228R, T240Y, G258R, C288Y, T306Y, C309T, T312K, G318R, G327R, C339Y, G345R, C348Y, C372Y, A387R, C390Y, C415Y, A432R, C438Y, C453M, G456R, T471Y, A489R, T519Y, C522S, C558Y, C561Y, C564Y, G576R, C579S	MT542770
B	–	10	AY072727	1–729	C165T, A183G	MT542771
		3	AY072727	93–753	C165Y, A183R	MT542772
		1	AY072727	97–593	C165Y, A183R, C204M, C309Y, C366Y, T519Y	MT542773
		1	AY072727	96–592	C165T, A183G, A272R, C406Y	MT542774
		1	AY072727	93–709	C165Y, A183R, C309Y	MT542775
		4	AY072727	97–753	C165Y, A183R, C309Y, T519Y	MT542776
		1	AY072727	93–600	C165Y, A183R, C309Y, T519Y, C543Y	MT542777
		2	AY072727	93–753	C165T, A183R, C309Y, C543Y	MT542778
		2	AY072727	97–753	C165Y, A183R, T519Y	MT542779
		1	AY072727	93–604	C165Y, C309Y, G354R, T519C	MT542780
		1	AY072727	97–719	C165Y, C309Y, G354R, T519Y, C543Y	MT542781
		1	AY072727	93–753	C165Y, C309T, G354R, T519Y, C543Y	MT542782
		1	AY072727	97–601	C165Y, C309T, T519Y	MT542783
		1	AY072727	93–711	C165Y, C309T, T519Y, C543Y	MT542784
		1	AY072727	97–701	C165Y, C309Y, C543Y	MT542785
		1	AY072727	97–719	C165Y, C309T, C543Y	MT542786
		1	AY072727	97–753	G180R, C309Y, T519C	MT542787
		1	AY072727	103–604	A183R	MT542788
		1	AY072727	98–714	A183R, C204M, T519Y	MT542789
		1	AY072727	97–706	C309Y	MT542790
		1	AY072727	102–604	C309T, T519C	MT542791
		1	AY072727	93–753	C309T, T519Y	MT542792
		1	AY072727	97–719	T519C	MT542793
		2	AY072727	1–753	T519Y	MT542794

K, G/T; M, A/C; R, A/G; Y, C/T.

**Table 7 pathogens-10-00206-t007:** Diversity, frequency, and main molecular features of *G. duodenalis* sequences at the *tpi* locus generated in the present study. GenBank accession numbers are provided.

Assemblage	Sub-Assemblage	No. of Isolates	Reference Sequence	Stretch	Single Nucleotide Polymorphisms	GenBank ID
A	AII	8	U57897	294–805	None	MT542795
		5	U57897	275–805	C287G	MT542796
		1	U57897	275–805	C287G, A381M	MT542797
	AII + BIII	1	U57897	289–754	94 SNPs	MT542798
		1	U57897	313–720	95 SNPs	MT542799
B	BIII	7	AF069561	1–456	None	MT542800
		1	AF069561	1–456	A8R, C108T, C111Y, C201Y	MT542801
		1	AF069561	1–456	A8R, C108Y, C201Y	MT542802
		2	AF069561	1–456	A10R	MT542803
		1	AF069561	1–456	A10R, C104Y, A372R	MT542804
		1	AF069561	1–456	A10R, C108Y, C111Y	MT542805
		1	AF069561	1–456	C15Y, C33Y, C34Y, G105R, C111Y, G254R	MT542806
		1	AF069561	1–456	G21A, C34T, C108T	MT542807
		1	AF069561	26–456	C34Y, G105R, T314Y	MT542808
		1	AF069561	1–456	G48A, G391A	MT542809
		1	AF069561	17–456	G105R, C108Y, C201Y, G391R,	MT542810
		1	AF069561	1–456	C108Y	MT542811
		1	AF069561	1–456	C108Y, C111Y	MT542812
		1	AF069561	38–456	C108Y, C111Y, C201Y	MT542813
		1	AF069561	1–456	C108Y, C201Y	MT542814
		1	AF069561	1–456	C108Y, G258R	MT542815
		2	AF069561	1–456	G198R, G207R	MT542816
	BIV	1	AF069560	1–479	A176G, A395G, C470T	MT542817
	BIII/BIV	1	AF069560	1–479	A5R, A33R, T57Y, C112Y, T131Y, T134Y, A176G, G281R, T314Y, A395G, C470Y	MT542818
		1	AF069560	1–479	A5R, A33R, T57Y, T131Y, T134Y, A176G, G281R, A395G, C470Y	MT542819
		1	AF069560	1–479	A5R, T57Y, T131Y, T134Y, A176G, A181R, A395G, C470T	MT542820
		1	AF069560	1–479	A5R, T57Y, T131Y, T134Y, A176G, G281R, A395G, A421M, C470Y	MT542821
		1	AF069560	1–479	A5R, A33R, T57Y, G128R, T131Y, T134Y, A176G, A395G, C470Y	MT542822
		2	AF069560	1–479	A5R, T57Y, G128R, T131Y, T134Y, A176G, A395G, C470Y	MT542823
		1	AF069560	1–479	A5R, T57Y, G128R, T131Y, T134Y, A176G, A395G	MT542824
		1	AF069560	1–479	A5R, T57Y, T131Y, T134Y, A176G, C237Y, A395G, C470Y,	MT542825
		1	AF069560	1–479	A5R, G44R, T57Y, C127Y, T131Y, T134C, A176G, A395R	MT542826
		1	AF069560	1–479	T57Y, T131Y, T134Y, A176G, G221R, G230R, A395G, C470Y	MT542827
		1	AF069560	44–479	T57Y, T131Y, T134Y, A176G, T317Y, A395G, C470Y	MT542828
		1	AF069560	1–479	T57Y, T131Y, T134Y, A176G, A395G, A437R, A449R, C470Y, G476R	MT542829

M, A/C; R, A/G; S, G/C; Y, C/T.

**Table 8 pathogens-10-00206-t008:** Univariable analysis. Crude association between *G. duodenalis* infections and variables of interest. *p*-values marked in bold indicate numbers that are significant on the 95% confidence limit (CI).

	*G. duodenalis* (Negative vs. Ever Positive)		Crude Association			
Variable	0, *n* = 366 ^1^	1, *n* = 198 ^1^	*n*	OR ^2^	95% CI ^2^	*p*-Value
Sex			564			
Female	177 (62%)	109 (38%)		—	—	
Male	189 (68%)	89 (32%)		0.76	0.54–1.08	0.13
Age group (years)			562			
0–4	38 (45%)	46 (55%)		—	—	
5–9	67 (57%)	50 (43%)		0.62	0.35–1.08	0.093
10–14	58 (62%)	35 (38%)		0.5	0.27–0.91	**0.023**
≥15	201 (75%)	67 (25%)		0.28	0.16–0.46	**<0.001**
Unknown	2	0				
Tribe			564			
1	155 (59%)	108 (41%)		—	—	
2	50 (73%)	19 (27%)		0.55	0.30–0.96	**0.041**
3	55 (80%)	14 (20%)		0.37	0.19–0.67	**0.002**
4	62 (85%)	11 (15%)		0.25	0.12–0.49	**<0.001**
5	20 (40%)	30 (60%)		2.15	1.17–4.04	**0.015**
6	24 (60%)	16 (40%)		0.96	0.48–1.87	0.9
Faecal consistency			564			
1	101 (68%)	48 (32%)		—	—	
2	200 (69%)	89 (31%)		0.94	0.61–1.44	0.8
3	9 (56%)	7 (44%)		1.64	0.56–4.65	0.4
4	56 (51%)	54 (49%)		2.03	1.22–3.38	**0.006**
Faecal appearance			564			
Normal	336 (65%)	182 (35%)		—	—	
Other (mucus, bloody)	30 (65%)	16 (35%)		0.98	0.51–1.83	>0.9
Abdominal pain			563			
No	175 (65%)	93 (35%)		—	—	
Yes	190 (64%)	105 (36%)		1.04	0.74–1.47	0.8
Unknown	1	0				
Vomit			564			
No	353 (65%)	192 (35%)		—	—	
Yes	13 (68%)	6 (32%)		0.85	0.29–2.18	0.7
Drinking water source			564			
River	1 (100%)	0 (0%)		—	—	
Well	1 (100%)	0 (0%)		1.00	0.00–36,409	>0.9
Piped	364 (65%)	198 (35%)		1,152,197	0.00–NA	>0.9
Treated water			564			
No	350 (65%)	190 (35%)		—	—	
Yes	16 (67%)	8 (33%)		0.92	0.37–2.13	0.9
Hand washing			564			
No	166 (59%)	115 (41%)		—	—	
Yes	200 (71%)	83 (29%)		0.60	0.42–0.85	**0.004**
Washing fresh produce			563			
No	77 (68%)	36 (32%)		—	—	
Yes	289 (64%)	161 (36%)		1.19	0.77–1.87	0.4
Unknown	0	1				
Eating with			563			
Hand	301 (64%)	167 (36%)		—	—	
Flatware	64 (67%)	31 (33%)		0.87	0.54–1.38	0.6
Unknown	1	0				
Defecation place			564			
Toilets	35 (69%)	16 (31%)		—	—	
Woods	294 (68%)	139 (32%)		1.03	0.56–1.98	>0.9
Yard	37 (46%)	43 (54%)		2.54	1.23–5.41	**0.013**
Contact with animals			564			
No	55 (63%)	32 (37%)		—	—	
Yes	311 (65%)	166 (35%)		0.92	0.57–1.49	0.7
Rotavirus			564			
No	365 (65%)	198 (35%)		—	—	
Yes	1 (100%)	0 (0%)		0.00		>0.9
*Ancylostoma*			564			
No	283 (64%)	162 (36%)		—	—	
Yes	83 (70%)	36 (30%)		0.76	0.49–.16	0.2
*Ascaris*			564			
No	364 (65%)	195 (35%)		—	—	
Yes	2 (40%)	3 (60%)		2.80	0.46–21.4	0.3
*Blastocystis*			564			
No	305 (64%)	173 (36%)		—	—	
Yes	61 (71%)	25 (29%)		0.72	0.43–1.18	0.2
*Chilomastix*			564			
No	307 (66%)	162 (34%)		—	—	
Yes	59 (62%)	36 (38%)		1.16	0.73–1.82	0.5
*E. coli*			564			
No	141 (59%)	97 (41%)		—	—	
Yes	225 (69%)	101 (31%)		0.65	0.46–0.93	**0.017**
*E. histolytica*			564			
No	216 (63%)	125 (37%)		—	—	
Yes	150 (67%)	73 (33%)		0.84	0.59–1.20	0.3
*E. nana*			564			
No	129 (59%)	88 (41%)		—	—	
Yes	237 (68%)	110 (32%)		0.68	0.48–0.97	**0.032**
*Hymenolepis*			564			
No	333 (65%)	180 (35%)		—	—	
Yes	33 (65%)	18 (35%)		1.01	0.54–1.82	>0.9
*Iodamoeba*			564			
No	336 (64%)	191 (36%)		—	—	
Yes	30 (81%)	7 (19%)		0.41	0.16–0.90	**0.038**
*Isospora*			564			
No	366 (65%)	197 (35%)		—	—	
Yes	0 (0%)	1 (100%)		1,447,714	0.00, NA	>0.9
*Sarcocystis*			564			
No	359 (65%)	196 (35%)		—	—	
Yes	7 (78%)	2 (22%)		0.52	0.08–2.19	0.4
*Strongyloides*			564			
No	345 (64%)	191 (36%)		—	—	
Yes	21 (75%)	7 (25%)		0.60	0.23–1.38	0.3
*Taenia*			564			
No	364 (65%)	198 (35%)		—	—	
Yes	2 (100%)	0 (0%)		0.00		>0.9
*Trichuris*			564			
No	365 (65%)	197 (35%)		—	—	
Yes	1 (50%)	1 (50%)		1.85	0.07–47.0	0.7
*Cyclospora*			564			
No	351 (65%)	188 (35%)		—	—	
Yes	15 (60%)	10 (40%)		1.24	0.53–2.80	0.6
No. of samples			564	1.72	1.43–2.08	**<0.001**
1	71 (83%)	15 (17%)				
2	99 (73%)	37 (27%)				
3	129 (64%)	72 (36%)				
4	67 (48%)	74 (53%)				

^1^ Statistics presented: *n* (%). ^2^ OR, crude odds ratio; CI, confidence interval; NA, not applicable.

**Table 9 pathogens-10-00206-t009:** Multivariable analysis comparing *G. duodenalis*-negative results versus *G. duodenalis* ever positive results. *p*-values marked in bold indicate numbers that are significant on the 95% confidence limit (CI).

Variable	aOR ^1^	95% CI ^1^	*p*-Value
Age group (years)			
0–4	—	—	
5–9	0.58	0.30–1.12	0.11
10–14	0.40	0.20–0.81	**0.011**
≥15	0.20	0.11–0.39	**<0.001**
Number_samples	1.46	1.18–1.81	**<0.001**
Tribe			
1	—	—	
2	0.46	0.24–0.85	**0.016**
3	0.43	0.21–0.85	**0.018**
4	0.31	0.14–0.63	**0.002**
5	1.83	0.94–3.60	0.075
6	0.94	0.45–1.95	0.9
Washing fresh produce			
No	—	—	
Yes	1.95	1.12–3.44	**0.020**
Faecal consistency			
1	—	—	
2	0.93	0.59–1.49	0.8
3	2.37	0.73–7.51	0.14
4	1.84	1.01–3.37	**0.046**
Faecal appearance			
Normal	—	—	
Other (mucus, bloody)	0.51	0.23–1.08	0.087

^1^ aOR, adjusted odds ratio; CI, confidence interval. *n* = 559 (removing five observations with missing values).

## Data Availability

All relevant data are within the article and its additional files. The sequences obtained in this study have been deposited in GenBank under accession numbers MT542718-MT542765 (*gdh*), MT542766-MT542794 (*bg*), and MT542795-MT542829 (*tpi*).

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
