# Peer review of "Multilocus Genotyping of Giardia duodenalis in Mostly Asymptomatic Indigenous People from the Tapirapé Tribe, Brazilian Amazon"

_pathogens, 2021, doi:10.3390/pathogens10020206_

Round 1
Reviewer 1 Report
Köster et al., have studied the distribution of Giardia genotypes in asymptomatic indigenous people in the Brazilian Amazon. This was done using multi-locus sequencing and analyses of patient data. It is a well-written paper but a few more analyses are needed.
Line 154. Table 4 and Methods. It is unclear how the genotype was assigned for each isolate. With this type of data collected in this project it could have been done in a more systematic way. First, make trees for all sub. Sequences (gdh, bg and tpi). Use good reference sequences from all major subgenotypes, see eg papers by
Soba B, Islamovic S, Skvarc M, Caccio SM. Folia Parasitol (Praha). 2015 Nov 6;62:2015.062
Ankarklev , Lebbad M, Petersson I, Karlsson L, Botero-Kleiven S, Andersson JO, Svenungsson B, Svärd SG.
PLoS Negl Trop Dis. 2011
Ankarklev J, Lebbad M, Einarsson E, Franzén O, Ahola H, Troell K, Svärd SG.
Infect Genet Evol. 2018 Jun;60:7-16.
This should make it possible to assign sub-genotypes for most. The three sequences should be catenated from all isolates where it is possible. These sequences can be added to a new tree in order to determine the sub-genotype. I suggest to use the G. muris sequences as out-groups.
Line 164, See papers above for better resolution. In the gdh tree presented in Fig 1 the strains can be both AI and AII strains but still they are annotated as AII. I suggest to make a separate A tree and a separate B tree since they have so different levels of substitutions, see Soba.
Line 178. Tables 5-7 can be put in Supp. Data.
Line 184. L40510 was originally isolated in Australia which is interesting.
Line 188. Many SNPs are found in positions where double peaks are seen in the sequencing. This is either mixed infections or allelic sequence divergence. What do the distribution look like in these positions when it comes to nucleotides? Is it more than 2 variants? Is it 50/50, 75/25? Do the changes change amino-acid or insert a stop codon?
Line 201. Compare with other data eg. sequences from Nicaragua, Lebbad 2008, https://doi.org/10.1016/j.actatropica.2008.01.004
Line 224. Again make trees with gdh, bgi and tpi and concatenated sequences. Use better reference sequences.
Line 327. Giardia might be part of the eukaryome instead in these asymptomatic patients. See eg paper by Stensvold et al, Pinning down the role of common luminal intestinal parasitic protists in human health and disease - status and challenges.
Stensvold CR. Parasitology. 2019 May;146(6):695-701.
Line 333. Was there any recreational use of water, eg swimming in the river by children. This is a known risk factor.
Line 410 See recent article about variability in assemblage B isolates,
Hidden Diversity within Common Protozoan Parasites Revealed by a Novel Genomotyping Scheme. Seabolt MH, Konstantinidis KT, Roellig DM.Appl Environ Microbiol. 2021 Jan 4:AEM.02275-20.
Also assemblage A, see Ankarklev J, Lebbad M, Einarsson E, Franzén O, Ahola H, Troell K, Svärd SG. Infect Genet Evol. 2018 Jun;60:7-16 but you need to look at more variable loci.
Author Response
Reviewer #1
Köster et al., have studied the distribution of Giardia genotypes in asymptomatic indigenous people in the Brazilian Amazon. This was done using multi-locus sequencing and analyses of patient data. It is a well-written paper but a few more analyses are needed.
We thank Reviewer #1 for his/her preliminary positive appraisal. We have attempted to thoroughly reply to all the comments and suggestions raised in his/her review (see below).
- Line 154. Table 4 and Methods. It is unclear how the genotype was assigned for each isolate. With this type of data collected in this project it could have been done in a more systematic way. First, make trees for all sub. Sequences (gdh, bg and tpi). Use good reference sequences from all major subgenotypes, see eg papers by
- Soba B, Islamovic S, Skvarc M, Caccio SM. Folia Parasitol (Praha). 2015 Nov 6;62:2015.062
- Ankarklev , Lebbad M, Petersson I, Karlsson L, Botero-Kleiven S, Andersson JO, Svenungsson B, Svärd SG. PLoS Negl Trop Dis. 2011
- Ankarklev J, Lebbad M, Einarsson E, Franzén O, Ahola H, Troell K, Svärd SG. Infect Genet Evol. 2018 Jun;60:7-16.
Reply: genotypes were assigned by direct comparison of the results obtained at the three loci (gdh, bg, and tpi) investigated. Results obtained at the gdh and tpi loci involving assemblage B sequences were prioritised over those obtained at the bg locus, as gdh and tpi (but not bg) allow genotyping at the sub-assemblage level and are therefore more informative. Sequences presenting double peaks that could not be clearly assigned to a given assemblage/sub-assemblage were expressed as ambiguous (e.g. AII/AIII or BIII/BIV) sequences. Mixed infections were only determined when two different assemblages were obtained in two or more of the loci investigated. This information has been now stated in lines 190-193 of the current manuscript.
Following Reviewer #1 advice, we have now prepared additional phylogenetic trees for sequences generated at the bg (Figure S1) and tpi (Figure S2) loci. This information has been presented as supplementary material as indicated in current lines 267 and 268. Appropriate reference sequences have been now included in the analyses. Please note that this new set of reference sequences is the one routinely used in our laboratory for phylogenetic and evolutionary analyses during the last 7 years, appearing in more than 30 publications.
- This should make it possible to assign sub-genotypes for most. The three sequences should be catenated from all isolates where it is possible. These sequences can be added to a new tree in order to determine the sub-genotype. I suggest to use the muris sequences as out-groups.
Reply: As requested by Reviewer #1, Figure 1 provides now a Maximum Parsimony and Bayesian phylogenetic tree based on gdh sequences including a new dataset of reference sequences. As demonstrated in the new analysis, no clear distinction can be done between BIII and BIV sequences within assemblage B, corroborating the results obtained in our previous version of this manuscript. Following Reviewer #1 advise, C. muris has been now added as outgroup.
- Line 164, See papers above for better resolution. In the gdh tree presented in Fig 1 the strains can be both AI and AII strains but still they are annotated as AII. I suggest to make a separate A tree and a separate B tree since they have so different levels of substitutions, see Soba.
Reply: Following reviewer #1 advice, we have split the phylogenetic tree to independently analyse assemblage A and assemblage B sequences (see inserted figures below). As you can see, the obtained results do not improve the resolution of the analyses conducted in current Figure 1, which has been, therefore, kept in the main body of the manuscript. The same region on gdh genes has the same selection pressure and different levels of substitutions for Assemblages cannot be assumed.
- Line 178. Tables 5-7 can be put in Supp. Data.
Reply: since this paper has already 12 supplementary tables and we have added in this new version two new supplementary figures and considering the relevance of the information provided in Tables 5-7, we prefer keeping them in the main body of the manuscript to facilitate their use as reference material for other molecular epidemiological studies. Please note that Pathogens has no restriction in the length of manuscripts, so in this particular situation we feel that this option is the most practical and convenient.
- Line 184. L40510 was originally isolated in Australia which is interesting.
Reply: No additional comments to this observation.
- Line 188. Many SNPs are found in positions where double peaks are seen in the sequencing. This is either mixed infections or allelic sequence divergence. What do the distribution look like in these positions when it comes to nucleotides? Is it more than 2 variants? Is it 50/50, 75/25? Do the changes change amino-acid or insert a stop codon?
Reply: All double peaks detected at chromatogram inspection involved two genetic variants at variable proportions even in the same sequence, making difficult to determine whether this effect was caused by a genetic recombination event or a true mixed infection. The occurrence of mixed infections can be investigated by cloning of PCR amplicons or next generation sequencing, but none of these approaches were used in the present study. Please note that this fact was already mentioned as a limitation of the study in current lines 465-466 of the manuscript.
Regarding the implication of double peaks in the potential occurrence of amino acid change at the protein level, we did not investigate further this possibility. However, since more of the double peaks detected involved R (A/G) or Y (C/T) single nucleotide polymorphisms associated to transition mutations we would expect a low rate of amino acid change at the protein chain.
- Line 201. Compare with other data eg. sequences from Nicaragua, Lebbad 2008, https://doi.org/10.1016/j.actatropica.2008.01.004
Reply: In order to keep the focus in the Brazilian scenario, we have preferred to include in our analysis sequences of human, animal, and environmental origin on this country only. Please note that there are hundreds of gdh sequences deposited in GenBank from countries other than Brazil, so it would be difficult to justify the presence of only few of them and no others.
- Line 224. Again make trees with gdh, bg and tpi and concatenated sequences. Use better reference sequences.
Reply: please see our answer to comment 2 by Reviewer #1. Phylogenetic trees for bg and tpi sequences have been now added as supplementary material (Figures S1 and Figure S2). This information is now provided in current lines 267 and 268.
The number of sequences available in GenBank are unequal for the different genes and could generate a low-resolution topology. Thus, we opted for separated analyses. In addition, all sequences from indigenous people remained in the same groups in the different analyses.
- Line 327. Giardia might be part of the eukaryome instead in these asymptomatic patients. See eg paper by Stensvold et al, Pinning down the role of common luminal intestinal parasitic protists in human health and disease - status and challenges. Stensvold CR. Parasitology. 2019 May;146(6):695-701.
Reply: although the finding that Giardia is not associated with or is perhaps even protective against acute diarrhoea was already proposed by Bartelt and Platts-Mills (see current reference #24), following Reviewer #1 advice we have also added the reference Stensvold 2015 to reinforce further this concept in current lines 369-373. Following the same line of reasoning, we have also introduced the concept of pathobiont to define those organisms that are usually commensal symbionts but under certain circumstances may act as true pathogens.
- Line 333. Was there any recreational use of water, eg swimming in the river by children. This is a known risk factor.
Reply: This possibility has been now acknowledged in current line 377 of the main body of the text.
- Line 410 See recent article about variability in assemblage B isolates, Hidden Diversity within Common Protozoan Parasites Revealed by a Novel Genomotyping Scheme. Seabolt MH, Konstantinidis KT, Roellig DM.Appl Environ Microbiol. 2021 Jan 4:AEM.02275-20. Also assemblage A, see Ankarklev J, Lebbad M, Einarsson E, Franzén O, Ahola H, Troell K, Svärd SG. Infect Genet Evol. 2018 Jun;60:7-16 but you need to look at more variable loci.
Reply: Reviewer #1 raises an important issue. The lack of resolution of current genotyping schemes, particularly for assemblage B sequences, has been now acknowledged in the paragraph devoted to the limitations of the study in current lines 472-477 of the main body of the text. The references Seabolt et al. 2021 and Ankarklev et al. 2018 have been included to illustrate this important point.

Reviewer 2 Report
This large-scale study on Giardia lamblia prevalence in members of the Tapirapé nation from central Brazil was extraordinarily well performed and offers plenty of interesting and useful information to the reader. Indeed, the study constitutes an imposing accomplishment and the presentation is equally sound. Consequently, I have no reservations about its publication whatsoever.
Only one very minor change regarding several tables in the paper would improve readability: instead of "isolate(s) no." it should read "no. of isolates" as the former term implies that a specific isolate is meant whereas the second refers to the number of isolates with a given trait.
Author Response
Reviewer #2
This large-scale study on Giardia lamblia prevalence in members of the Tapirapé nation from central Brazil was extraordinarily well performed and offers plenty of interesting and useful information to the reader. Indeed, the study constitutes an imposing accomplishment and the presentation is equally sound. Consequently, I have no reservations about its publication whatsoever.
We thank Reviewer #2 for his/her positive appraisal.
- Only one very minor change regarding several tables in the paper would improve readability: instead of "isolate(s) no." it should read "no. of isolates" as the former term implies that a specific isolate is meant whereas the second refers to the number of isolates with a given trait.
Reply: following Reviewer #2 recommendation, the requested change has been introduced in current Tables 5-7.
Reviewer 3 Report
Dear Authors,
Your manuscript describes important epidemiological aspect of spread of Giardia duodenalis in indigenous people from the Tapirapé tribe, Brazilian Amazon. It is important to keep track not only on well known parasites with high virulence, but also on the parasites that have relatively low pathogenicity.
The manuscript is carefully prepared. The results are described in details and are easy to follow and to understand. Methods can be repeated based on the provided protocols in Materials and Methods section.
I recommend to accept the manuscript.
Author Response
Reviewer #3
Your manuscript describes important epidemiological aspect of spread of Giardia duodenalis in indigenous people from the Tapirapé tribe, Brazilian Amazon. It is important to keep track not only on well-known parasites with high virulence, but also on the parasites that have relatively low pathogenicity.
The manuscript is carefully prepared. The results are described in detail and are easy to follow and to understand. Methods can be repeated based on the provided protocols in Materials and Methods section.
I recommend accepting the manuscript.
We thank Reviewer #3 for his/her positive appraisal.
Round 2
Reviewer 1 Report
The paper has been revised and it is now acceptable for publication.